# Combined Polarization/Magnetic Modulation of a Transverse NMR Gyroscope

**DOI:** 10.3390/s23104649

**Published:** 2023-05-11

**Authors:** Susan S. Sorensen, Thad G. Walker

**Affiliations:** Department of Physics, University of Wisconsin-Madison, Madison, WI 53706, USA

**Keywords:** NMR, SEOP, gyroscope

## Abstract

In this paper, we describe a new approach to the continuous operation of a transverse spin-exchange optically pumped NMR gyroscope that utilizes modulation of both the applied bias field and the optical pumping. We demonstrate the simultaneous, continuous excitation of ${}^{131}$Xe and ${}^{129}$Xe using this hybrid modulation approach and the real-time demodulation of the Xe precession using a custom least-squares fitting algorithm. We present rotation rate measurements with this device, with a common field suppression factor of ∼1400, an angle random walk of 21 μHz/$\sqrt{\mathrm{Hz}}$, and a bias instability of ∼480 nHz after ∼1000 s.

## 1. Introduction

Spin-exchange (SE)-pumped nuclear magnetic resonance (NMR) gyroscopes [1,2,3,4,5,6] measure nonmagnetic spin-dependent interactions [7,8,9,10,11,12,13,14,15,16,17], such as inertial rotation [1,18], by monitoring deviations in the precession of spin-polarized nuclei about an applied bias magnetic field.

SE-pumped NMR comagnetometers have been of special interest in the precision-measurement community because of their potential for miniaturization, especially when compared to the size scaling of similar devices, such as ring laser gyroscopes [19]. Continuous operation is preferable for applications such as inertial navigation. The transverse NMR gyroscope presented in this work was developed to allow for continuous operation, while also suppressing systematic errors from longitudinal polarization. One other approach to continuous drive is operation about a compensation point [2,18,20]. The scale factor (also known as the conversion factor between the measured signals and the desired output) of these self-compensating devices depends on experimental factors and, therefore, must be calibrated. In comparison, the transverse NMR gyroscope has a scale factor that depends only on known physics constants.

In our previously presented works on the transverse NMR gyroscope, we demonstrated continuous operation and suppression of common systematic errors [21]. We showed how the precession of two noble gas species can be continuously excited using polarization modulation [22], or using modulations of the pulsed bias field [4]. In this work, we present a transverse NMR gyroscope that excites Xe using a new hybrid approach, combining both polarization modulation and pulse density modulation. The hybrid approach may allow us to take advantage of the benefits of both polarization modulation (PM) and pulse density modulation (PDM) to potentially improve signal detection and/or systematic errors.

We describe the fundamentals of operation for a hybrid-driven NMR gyroscope in Section 2. In Section 3, we derive the Xe and Rb polarizations using the Bloch equations, and we show how we use the results of this derivation to design our detection and least-squares demodulation. Then, in Section 4, we present experimental results of comagnetometry under three conditions: open loop, magnetic field feedback, and combined field and drive frequency feedback. We also discuss systematic uncertainties of the experiment.

## 2. Device Overview

A simplified schematic is shown in Figure 1. A vapor cell contains noble gas species and alkali-metal atoms [23], as well as a buffer gas. For the system described in this paper, the noble gas species were ${}^{129}$Xe and ${}^{131}$Xe and the alkali-metal was ${}^{85}$Rb. The Rb atoms were optically pumped using circularly polarized lasers propagating in the $\hat{x}$ direction. During collisions between the gaseous Xe and Rb, a hyperfine interaction transfers spin between the two, as has been extensively studied [24,25,26]. The Xe nuclei were thereby spin-polarized through spin-exchange collisions with the Rb electrons.

We applied a (time-dependent) bias magnetic field along $\hat{z}$.

Since the spin polarization was transverse to the pumping direction, the Xe spun precess about the applied magnetic field. In order to continuously polarize the nuclei, we performed Bell–Bloom pumping [27] in order to synchronize the precession of the electrons and nuclear spins [21]. In this approach, the synchronization was accomplished by applying the bias magnetic field as a sequence of short, low-duty-cycle pulses, with each pulse tuned to produce a 2π radian rotation of the Rb [28]. The Rb atoms behaved as if they were in a zero field. In contrast, the Xe nuclei responded only to the (nonzero) time average of the bias field pulses. Thus, the Xe nuclei precessed freely about the average bias field, while the Rb atoms were stably polarized along the pumping direction [21]. By applying modulations to the field and/or optical pumping, the Xe nuclei could be excited continuously [3,4,22].

The pulsed field was applied using two pairs of square Helmholtz coils with differing side lengths, wound in series with opposite polarity. This design minimized field gradients over the area of the cell and limited coupling to the end caps of the three-layer cylindrical μ-magnetic shield that housed our cell and coils. These coils were custom-designed and wrapped on a 3D printed ABS plastic rig. A custom-made H-bridge circuit [29] was used to apply short (<5 μs) pulses of ∼1 ampere peak current. The pulses were triggered by a signal supplied by an FPGA, which provided precise control of the pulse timing.

The optical apparatus used for this work is shown in Figure 2. It was a modified version of the apparatus described in detail in [29] for polarization modulation operation, with the primary difference being the addition of the quarter wave plates (QWPs), which are labeled as B. Modulation of the circularly polarized pump light is accomplished using DC-coupled electro-optic modulators, which are configured to output right- or left-hand circularly polarized light. QWP A then transformed this to s- and p-linear polarized light to reduce polarization distortions upon reflection off the dichroic mirrors, and QWP B transformed the light back to circularly polarized just before the final steering lens and the cell.

The Rb atoms were also used as an in situ magnetometer to monitor the precession of the Xe. The Xe nuclei precessed in the transverse (x–y) plane. Once the Xe was out of axis with the Rb, the polarized Xe exerted a torque that tipped the Rb polarization slightly into the $\hat{z}$ direction. The z-component of the Rb polarization was monitored using a linearly polarized probe laser along $\hat{z}$ and a balanced Faraday detector [30,31]. A major advantage of NMR detection using an in situ magnetometer was a 500× enhancement of the signal size as compared to an external pickup loop [32,33].

The NMR frequency, Ω, of polarized Xe about a magnetic field, *B*, is given by
(1)Ω=γB+sX,
where γ is the gyromagnetic ratio, s={−1,1} encodes the sign of γ (such that Ω,γ>0), and *X* is generally any nonmagnetic spin-dependent interaction [34]. For a gyroscope, *X* is the rotation frequency $\Omega^{R}$. The magnetic field, *B*, includes contributions from the applied pulsed bias field ($B_{p}$), external magnetic fields and applied DC nulling fields ($B_{0}$), the Rb spin-exchange field experienced by the Xe $b_{S}^{K}S$ (where $b_{S}^{K}$ is the spin-exchange coefficient characterizing the influence of the Rb polarization on the Xe, see [21]), and any other applied fields. In order to suppress the effect of the Rb spin-exchange field on the Xe precession, we applied a “compensation” field [21], which was antiparallel to the Rb polarization to cancel $b_{S}^{K}S$.

By simultaneously measuring the precession of two colocated Xe species of known γ, we could eliminate the effect of the common field and deduce $\Omega^{R}$. This method of removing common field effects is referred to as comagnetometry. For the two Xe species in our system, let superscript *a* represent ${}^{129}$Xe and superscript *b* represent ${}^{131}$Xe. Then, $s^{a}=-1$ and $s^{b}=1$. Equation (Equation 1) is then written as
(2)$$\Omega^{a}=\gamma^{a}B+s^{a}\Omega^{R}$$
for species *a*, with an analogous equation for species *b*. In order to perform comagnetometry, we solved the system of equations to eliminate *B*, such that
(3)$$\frac{\rho \Omega^{b}-\Omega^{a}}{1+\rho }=\Omega^{R},$$
where $\rho =\gamma^{a}/\gamma^{b}=3.373417\left(38\right)$ [35].

## 3. Drive and Demodulation

### 3.1. Bloch Equations

The spin dynamics for the Xe nuclear polarization (K) and the Rb electron polarization (S) are described by the Bloch equations. The equation for the transverse Xe polarization for species *a* (with analogous equation for species *b*) can be written as
(4)$$\frac{dK_{+}^{a}}{dt}=-\left(s^{a}i\Omega_{z}^{a}+\Gamma_{2}^{a}\right)K_{+}^{a}+\Gamma_{S}^{a}S_{+}+s^{a}i\Omega_{+}^{a}K_{z}^{a},$$
where Ω is the Xe resonance frequency as given in Equation (Equation 1), $\Gamma_{2}$ is the transverse relaxation rate, and $\Gamma_{S}^{a}$ is the spin-exchange rate constant.

From Equation (Equation 4), we see that Xe can be continuously excited by modulating $\Omega_{+}$ [36], $S_{+}$, and/or $\Omega_{z}$. Since our transverse geometry resulted in negligible $K_{z}$, we did not modulate $\Omega_{+}$ in this work. We modulated $S_{+}$ by periodically reversing the polarization of the pump light that polarized the Rb, which we refer to as polarization modulation (PM). Since our bias z-field was applied as a series of pulses, we modulated $\Omega_{z}$ by changing the repetition frequency of the pulses, which we refer to as pulse density modulation (PDM). The Bloch equation derivations for a transverse NMR gyroscope driven with either PM or PDM can be found in [22] or [29], respectively. In this paper, we present hybrid modulation, which utilizes a combination of PM and PDM. The derivation for hybrid modulation is essentially a combination of the PM and PDM derivations and, therefore, only the main results are given here.

Our modulations were applied as follows. For PM, the pumping rate, R(t) was modulated along $\hat{x}$ as a square wave (the sign of a cosine), such that
(5)$$R\left(t\right)=R\;\mathrm{sign}\left(\cos\omega_{PM}t\right),$$
where $\omega_{PM}$ is the frequency of the applied modulation. For PDM, the repetition frequency of the applied bias pulses was modulated. Due to the slow response of the Xe (compared to the Rb), the short pulses applied at a rate of $\omega_{p}$ resulted in an effective field, $B_{p}=\omega_{p}/\gamma^{S}$, experienced by the Xe. The Rb was effectively insensitive to this field since each pulse resulted in a 2π Rb rotation [28]. We modulated $\omega_{p}$ sinusoidally, resulting in the Xe nuclei experiencing a time-averaged pulsed z-field
(6)$$B_{p}=B_{p0}\left(1+b_{1}\cos\left(\omega_{PDM}t\right)\right),$$
where $B_{p0}$ is the average field, $b_{1}$ is the modulation index, and $\omega_{PDM}$ is the frequency of the applied modulation. These modulations are shown in Figure 3.

Plugging these modulations into the Bloch equation, we obtain a resonance condition for exciting ${}^{129}Xe$ (with an analogous equation for ${}^{131}Xe$) in a hybrid drive scheme
(7)$$\omega_{d}^{a}=p^{a}\omega_{PM}+q^{a}\omega_{PDM},$$
where $\omega_{d}$ is the drive frequency of the Xe, *p* and *q* are integers, and *p* is odd. In principle, the NMR can be driven with any combinations of *p* and *q* that satisfy this equation for both isotopes. Our choice of *p* and *q* was informed by the desire to maximize our simultaneous Xe polarizations. From the steady state solution to the Xe Bloch equation, we know that the magnitude of the transverse Xe polarization, $K_{⊥}$, is proportional to the product of the Fourier coefficients from the two modulations, such that for species *a* (and similarly for species *b*)
(8)$$K_{⊥}^{a}=\frac{\Gamma_{S}^{a}}{\Gamma^{\prime }\Gamma_{2}^{a}}\frac{2}{p^{a}\pi }J_{q^{a}}\left(\frac{\omega_{d}^{a}b_{1}}{\omega_{PDM}}\right),$$
where $\Gamma^{\prime }$ is the magnetic width of the Rb magnetometer, the factor 2/pπ comes from the Fourier decomposition of the polarization modulation, and $J_{q}$ (the *q*th Bessel function of the first kind) arises from the sinusoidal modulation of the bias field (through the Jacobi–Anger identity, see [4]). We define an amplitude coefficient $j_{pq}^{a}=\frac{2}{p^{a}\pi }J_{q^{a}}\left(\frac{\gamma^{a}B_{p0}b_{1}}{\omega_{PDM}}\right)$ (with an analogous result for $j^{b}$). In general, $j^{a}$ and $j^{b}$ do not maximize at the same modulation index, but we looked for a scheme that gave a large *j* for both species at the same $b_{1}$. Furthermore, we required $0<b_{1}<1$ so that we did not need to switch the direction of the applied z-pulses. Figure 4 shows *j* versus $b_{1}$ for our chosen drive scheme, $\left(p^{a},q^{a}\right)=\left(1,1\right)$ and $\left(p^{b},q^{b}\right)=\left(1,-1\right)$. In this work, we used $b_{1}=0.78$, which approximately maximized the sum of the amplitude coefficients. We also chose this drive scheme to enable the dual species, dual output feedback scheme, which is discussed in Section 4.3.

### 3.2. Detection and Demodulation

In order to monitor the Xe precession, we used a z-probe laser and a balanced Faraday detector to measure a signal proportional to $S_{z}$. From the Bloch equation derivation we get the following:(9)Sz=−R(t)Γ′2(ΩyS−ΩzSΓ′ΩxS)=−R(t)Γ′2Im[γSbKSK+e−iϵz],
where we define
(10)$$\epsilon_{z}=\tan^{-1}\left(\gamma^{S}B_{z0}/\Gamma^{\prime }\right)$$
as the magnetometer phase shift that results from low frequency z-fields ($B_{z0}$), as discussed in [4]. We make the substitution $K_{+}=K_{⊥}e^{-si\phi }$ where $K_{⊥}$ and ϕ are the amplitude and phase of the transverse Xe precession, respectively. Then, we can see that the imaginary term in Equation (Equation 9) will include the term $-\sin(s^{a}\phi^{a}+\epsilon_{z})$, which can be re-written as $-s\sin(\phi^{a}+s^{a}\epsilon_{z})$ (plus analogous terms for species *b*). We then make the substitutions $\phi^{a,b}=\delta^{a,b}+\alpha^{a,b}$. The variable
(11)$$\alpha =\int (\omega_{d}+\gamma b_{1}B_{p0}\cos\left(\omega_{PDM}t\right))dt$$
is the expected Xe precession phase if the only field present is an ideal pulsed bias field and the resonance is driven perfectly. Note that α has the same form as in the pure PDM case [3], but with one fewer field modulation. The variable δ is the Xe precession phase shift (also known as how far the measured precession deviates from the expected phase α). Our measured $S_{z}$ is now of the form
(12)$$\begin{array}{ccc} S_{z}= & & [A^{S}\mathrm{sign}\left(\cos\left(\theta_{PM}\right)\right)+C^{S}]\times \\ & & [A^{a}\sin\left(\alpha^{a}\right)\cos\left(\delta^{a}-\epsilon_{z}\right)+A^{a}\cos\left(\alpha^{a}\right)\sin\left(\delta^{a}-\epsilon_{z}\right)\\ & & +A^{b}\sin\left(\alpha^{b}\right)\cos\left(\delta^{b}+\epsilon_{z}\right)+A^{b}\cos\left(\alpha^{b}\right)\sin\left(\delta^{b}+\epsilon_{z}\right)\\ & & +C^{y}]+C^{PD}, \end{array}$$
where $\theta_{PM}$ is $\omega_{PM}t$ plus an arbitrary phase offset, $A^{S,a,b}$ are amplitude constants, and $C^{S,y,PD}$ are DC offsets. If our experimental apparatus was perfect, all of the C constants would be zero, but, alas, we must allow for imperfections. Any asymmetry in the polarization modulation is represented as $C^{S}$, $C^{y}$ is from any DC y-fields, and $C^{PD}$ is a detection offset (e.g., from photodiode misbalance).

Figure 5 shows our recorded Faraday signal (dots) and our model waveform from Equation (Equation 12) (lines). In order to see how this signal is a combination of the precession of two Xe species, we can plot this signal in ways that account for our modulations. First, we multiplied by the PM modulation square wave to obtain a “rectified” signal. Then, we plotted the rectified signal vs. the pulse number (rather than time) to account for the pulse density modulation. We were left with a plot that is clearly a sum of two sine waves.

When we were demodulating pure PM signals [22], we used two single-frequency lock-in detectors and detected at one of several large discrete sideband frequencies. With the addition of PDM, however, the Xe frequencies needed to be constantly changed so that the signal was spread continuously over a large band of frequencies. There was no single frequency we could demodulate at that would give large signal content. Further, any single frequency that showed reasonable signal content from one Xe species would also show signal from the other Xe species due to the overlap in their frequency ranges, meaning that we could not isolate the signal from a single Xe species. Therefore, we designed a custom demodulation waveform. This demodulation scheme was developed from the idea that the Xe precessed a fixed amount during each applied pulse, no matter what the pulse repetition rate was, and we knew when each pulse was applied. Therefore, we knew what the expected Xe precession (α) was at any time. From there, we decided to use a least-squares fit for our demodulation [37].

We used the known form of our measured signal to design our custom fitting function. Equation (Equation 12) can be re-written as a matrix equation (see Appendix A) of the form $S_{z}=M.b$, where $S_{z}$ is measured, *M* is a matrix where each row contains all of our various known functions and variables (including the sine and cosine of $\alpha^{a,b}$ and $\mathrm{sign}(\cos\left(\theta_{PM}\right))$) evaluated at a specific time, and *b* is a column matrix of our unknowns (including the *A*s, *C*s, $\delta^{a,b}$, and $\epsilon_{z}$). This can be solves for *b* by matrix inversion, so $b={\left(M^{t}M\right)}^{-1}M^{t}S_{z}$. We measured 50 points before calculating *b*, such that $S_{z}$ was 50 × 1 and *M* was 50 × 10. Given our gated (see [4]) sampling frequency of 200 Hz, this resulted in an effective data acquisition rate of 4 Hz.

Figure 6 demonstrates the simultaneous ${}^{129}$Xe-${}^{131}$Xe excitation using hybrid drive and no feedback. We measured linewidths of 14 mHz and 16 mHz for ${}^{129}$Xe and ${}^{131}$Xe, respectively, with peak field sizes of 26 μG for ${}^{129}$Xe and 11 μG for ${}^{131}$Xe.

These linewidths were measured simultaneously by scanning the center pulsing frequency (thus changing the resonance frequencies) without changing the drive frequencies. The pulsing frequency was converted to an effective Xe Larmor frequency and plotted on the bottom axis. From the open loop phase noise measurements on the resonance, we found that the signal-to-noise ratio (SNR) was 2900 $\sqrt{\mathrm{Hz}}$ for ${}^{129}$Xe and 1100 $\sqrt{\mathrm{Hz}}$ for ${}^{131}$Xe.

## 4. Comagnetometry

We performed comagnetometry using our measured signals with the goal of obtaining an $\Omega^{R}$ that was independent of magnetic fields. There are a few different approaches to this, depending on how the gyroscope is run. In this section, we show how Equation (Equation 3) is implemented in practice for open loop, single feedback, and dual feedback operations.

### 4.1. Open Loop Comagnetometry

From Equation 12, we see that our demodulation did not give us the Xe precession phase shift $\delta^{a,b}$ directly, but rather the term $\delta^{a,b}+s^{a,b}\epsilon_{z}$. The Bloch equation gives an expression for $\dot{\delta }$, which we converted into the Fourier domain (such that $\tilde{f}=f\left(\omega \right)$) and solved to obtain
(13)$$\tilde{\delta^{a}}+s^{a}\tilde{\epsilon_{z}}=\frac{-\tilde{\omega_{d}^{a}}+\gamma^{a}\left({\tilde{B}}_{z0}+{\tilde{B}}_{p0}\right)+s^{a}{\tilde{\Omega }}^{R}+s^{a}i\omega {\tilde{\epsilon }}_{z}}{i\omega +\Gamma_{2}^{a}}$$
for species *a*, with an analogous expression for species *b*.

We used our measurements of $\delta^{a}-\epsilon_{z}$ and $\delta^{b}+\epsilon_{z}$ (as given by Equation (Equation 13)) to obtain a comagnetometer signal, which eliminated γB. For fixed drive (${\tilde{\omega }}_{d}\to 0$),
(14)$$\frac{\rho \left(\tilde{\delta^{b}}+\tilde{\epsilon_{z}}\right)\left(i\omega +\Gamma_{2}^{b}\right)-\left(\tilde{\delta^{a}}-\tilde{\epsilon_{z}}\right)\left(i\omega +\Gamma_{2}^{a}\right)}{1+\rho }=\tilde{\Omega^{R}}+i\omega \tilde{\epsilon_{z}}.$$

One drawback of this method of comagnetometry is that it requires knowledge of $\Gamma_{2}$, which is measure separately from the rotation measurements and can drift over time. Figure 7 shows the modified Allan deviation (MDEV) [38,39] for this comagnetometry calculation using open loop Xe measurements.

We have seen in [4] that $i\omega \tilde{\epsilon_{z}}$ is non-negligible, so in order to obtain a measure of only $\tilde{\Omega^{R}}$ we must either measure $i\omega \tilde{\epsilon_{z}}$ and include it in our analysis, or hold it to zero. In the following section, we chose to hold $i\omega \tilde{\epsilon_{z}}$ to zero using feedback [40]. We show that holding $i\omega \tilde{\epsilon_{z}}=0$ leads to a dramatic improvement in the bias instability.

### 4.2. Field Feedback

The magnetometer phase shift, $\epsilon_{z}$, depends on z-field drifts, as shown in Equation (Equation 10). Therefore, if we stabilize the z-fields, we can hold $\epsilon_{z}$ constant (thereby holding $i\omega \tilde{\epsilon_{z}}$ to zero). We applied a z-field correction, $B_{FB}$, such that
(15)$${\tilde{B}}_{FB}=\frac{1}{\gamma^{a}+\gamma^{b}}\times \left[\tilde{G^{a}}\left(\tilde{\delta^{a}}-\tilde{\epsilon_{z}}\right)+\tilde{G^{b}}\left(\tilde{\delta^{b}}+\tilde{\epsilon_{z}}\right)\right].$$

The gains, $G^{a,b}$, were calculated as integral gain with an inverted zero. This shape was chosen to give high gain at low frequency and avoid feedback instability from the phase shift at $\Gamma_{2}$. By using a weighted sum of the two Xe signals, we obtained an error signal that was independent of both rotation and $\epsilon_{z}$. Since we used two signals to apply a single correction, we refer to this as dual-species single output (DSSO) feedback. The field experienced by the Rb (which does not include the pulses) was then $B_{z0}^{\prime }=B_{z0}+B_{FB}$. In the high gain limit (G→∞), ${\tilde{B}}_{FB}\to -{\tilde{B}}_{z0}$ such that $B_{z0}^{\prime }$ goes to zero. When $B_{z0}^{\prime }$ is held to zero, ${\tilde{\epsilon }}_{z}$ is a held constant.

Having held $\tilde{\epsilon_{z}}$ constant and $i\omega \tilde{\epsilon_{z}}$ to zero using field feedback, our comagnetometry calculation from Equation (Equation 14) becomes
(16)$$\frac{\rho \tilde{\delta^{b}}\left(i\omega +\Gamma_{2}^{b}\right)-\tilde{\delta^{a}}\left(i\omega +\Gamma_{2}^{a}\right)}{1+\rho }=\tilde{\Omega^{R}}.$$

Thereby, DSSO field feedback allows us to deduce rotation independent of z-fields and magnetometer phase shifts.

Figure 8 demonstrates that the DSSO field feedback resulted in substantial suppression of the common magnetic field noise. By applying an ancillary AC, $B_{app}$, along $\hat{z}$, we could see how much that field was suppressed in our final measure of the rotation. For this test, we applied $B_{app}$ at 5 mHz with an amplitude of just over 200 μG. Since our final measure of rotation was in units of Hz, we cast our applied field in terms of the frequency noise it would cause on a single species of Xe. For this calculation, we used the gyromagnetic ratio of the less magnetically sensitive species, ${}^{131}$Xe. Our applied field would, therefore, be expected to produce a frequency oscillation, $\gamma^{b}*B_{app}$. We applied DSSO feedback and measured the resulting $\delta +s\epsilon_{z}$. If we used the transfer function to convert the ${}^{131}$Xe phase shift to frequency, we could see that the feedback alone suppressed the applied signal by a factor of 330. We then used our measured ${}^{129}$Xe signal to perform comagnetometry and deduce rotation. We saw that the oscillation on the rotation was a factor of 1400 smaller than the applied oscillation. We refer to this factor as the field suppression factor (FSF). Since this factor quantifies how much field noise can be expected on a measurement of rotation, we believe FSF is an important quantity to report when presenting results of an NMR gyroscope.

Applying our DSSO FB and taking rotation measurements, we found that our bias instability was very dependent on our DC transverse fields. Our rotation measurements showed an apparent dependence on transverse fields that go by the product $B_{x}B_{y}$. If $B_{x}$ or $B_{y}$ were nonzero, we had increased sensitivity to field noise along the other transverse direction. In order to find the optimal settings, we took three sets of measurements, one with varying DC $B_{y}$, one with varying DC $B_{x}$, and one with varying AC x-compensation field amplitude [21], as shown in Figure 9. We compared the MDEVs to find the transverse field settings that gave the lowest bias instability.

We repeated these scans every time we wanted to take a stability measurement, as low-frequency field drifts caused the optimal settings to drift over time. Indeed, we found that long measurements (even those taken after scanning the transverse fields) showed variation over time. For example, we took a 10-hour measurement that did not even reach a bias instability of 1 μHz. We cut this measurement into segments that were 5 ks in length and analyzed the segments individually. As shown in Figure 10, the stability of the magnetometer varied widely over the course of the long measurement. This meant that our measurements were not perfectly repeatable, but rather depended on how far the transverse fields drifted over the course of a given measurement. In the future, we will need to find a way to stabilize—or reduce the sensitivity to—transverse fields.

Figure 11 shows our best-to-date results for both hybrid drive and pure PDM (no PM). The PDM data was obtained with the same apparatus as the hybrid data. The apparatus was able to be easily switched between PDM and hybrid operation by inserting/removing two quarter wave plates and changing the modulations applied to the pulsed field and optical pumping. Both measurements also followed the same matrix inversion least-squares fit principle for demodulation. For hybrid drive, we found an angle random walk (ARW) of 21 $\mu \mathrm{Hz}/\sqrt{\mathrm{Hz}}$ and a bias instability of 480 nHz after about 1000 s. Comparing the two MDEVs, we saw similar stability performance for PDM and hybrid drive. In both cases, bias instability improvements were presented over our previously published results [3]. At this time, there is not an obvious benefit to operating with hybrid drive rather than pure PDM. It seems likely that both hybrid and PDM drive share the same limiting noise source. We suspect that transverse field noise limited the stability of our rotation measurements in both cases. In the future, when we can suppress transverse field noise (or reduce our sensitivity to it), we will revisit both hybrid and pure PDM drive to evaluate the best drive scheme moving forward.

### 4.3. Dual Output Feedback

Performing comagnetometry as given in Equation (Equation 16) still has the drawback of requiring us to know $\Gamma_{2}$. Since we do not measure $\Gamma_{2}$ simultaneously with our measures of $\delta +s\epsilon_{z}$, any noise on $\Gamma_{2}$ is mapped onto the rotation. We can theoretically circumvent this drawback using a second feedback loop to adjust the drive frequencies and hold the measured phase shifts equal to zero.

The second feedback loop uses a weighted difference of the two Xe signals to obtain an error signal (as opposed to the weighted sum used for the field feedback). Due to our choice of drive scheme, we can apply this correction to our drive frequencies by adjusting $\omega_{PDM}$, while keeping $\omega_{PM}$ fixed. The new feedback expressions for dual-species dual output (DSDO) feedback are then
(17)$$\begin{array}{c} \gamma^{a}B_{FB}-\omega_{PDM}=G^{a}\left(\delta^{a}-\epsilon_{z}\right),\\ \gamma^{b}B_{FB}+\omega_{PDM}=G^{b}\left(\delta^{b}+\epsilon_{z}\right). \end{array}$$

When these two feedback loops are used together, the total field noise is still held to zero in the high gain limit, and we also obtain that ${\tilde{\omega }}_{PDM}\to -{\tilde{\Omega }}^{R}$. Thus, we are able to directly obtain a measure of rotation noise without using $\Gamma_{2}$ in the calculations.

When we applied the DSDO feedback, we saw long timescale (1000 s) drifts. For reasons we don’t understand, rather than holding the drive frequencies to the Xe resonance, the feedback held the detuning to a point that started on resonance and then slowly drifted off resonance. If the feedback was holding our drive frequencies on resonance, the measured Xe amplitudes would remain constant. Instead, we observed that the measured Xe amplitude decreased over time, though the measured $\delta +s\epsilon_{z}$ was still held to zero. The effect was most dramatic on ${}^{129}$Xe. Figure 12 shows the amplitude drift on ${}^{129}$Xe during DSDO operation. We have seen similar drifting problems using various drive and feedback schemes, even when using pure PDM (rather than hybrid) drive. Comparing many instances of drift over the course of several years, the drift seems to be present when the following two conditions are met: (1) we are driving either Xe species using a combination of two modulation frequencies, and (2) we are applying feedback to a modulation frequency. Since the drift issue seems to have something to do with drive and/or modulation frequencies, and since all of our frequencies and feedback are calculated on an FPGA, we suspect there may be some kind of accumulated rounding error in our code. While our investigation is ongoing, we hope to in the future identify and eliminate the source of drift, allowing us to use DSDO feedback to further improve our comagnetometry calculation of rotation.

## Figures and Tables

**Figure 1 sensors-23-04649-f001:**
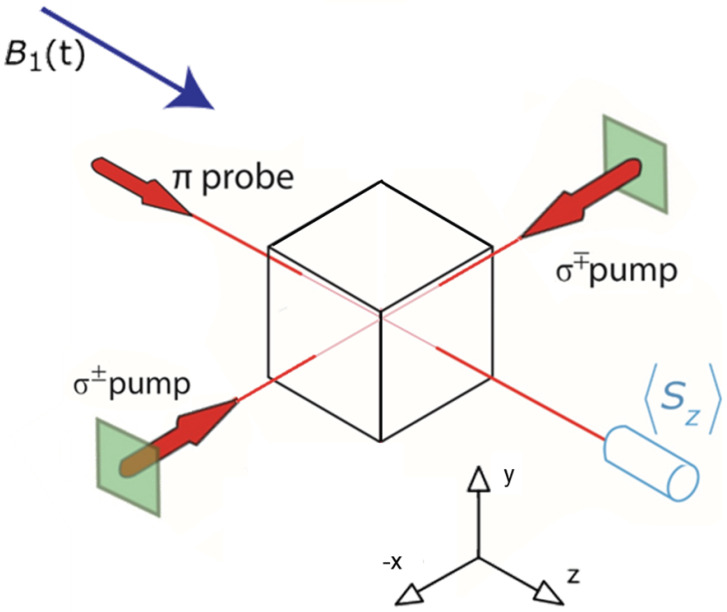
A simplified schematic of a transversely pumped NMR gyroscope. A cubic vapor cell contains ${}^{131}$Xe, ${}^{129}$Xe, and Rb. Circularly polarized pump lasers optically pump Rb electrons along $\hat{x}$. A linearly polarized probe laser propagates along $\hat{z}$. A pulsed bias magnetic field is applied along $\hat{z}$.

**Figure 2 sensors-23-04649-f002:**
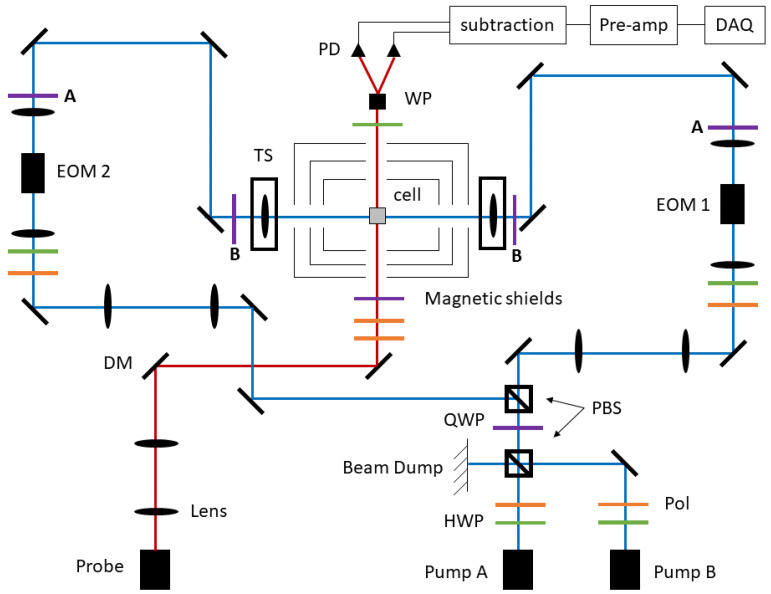
Diagram of the optical apparatus (not to scale). DM: dichroic mirror, Pol: linear polarizer, HWP: half wave plate, QWP: quarter wave plate, PBS: polarizing beam splitter, WP: Wollaston prism, PD: photodiode, EOM: electro-optic modulator, TS: two-axis translation stage with lens.

**Figure 3 sensors-23-04649-f003:**
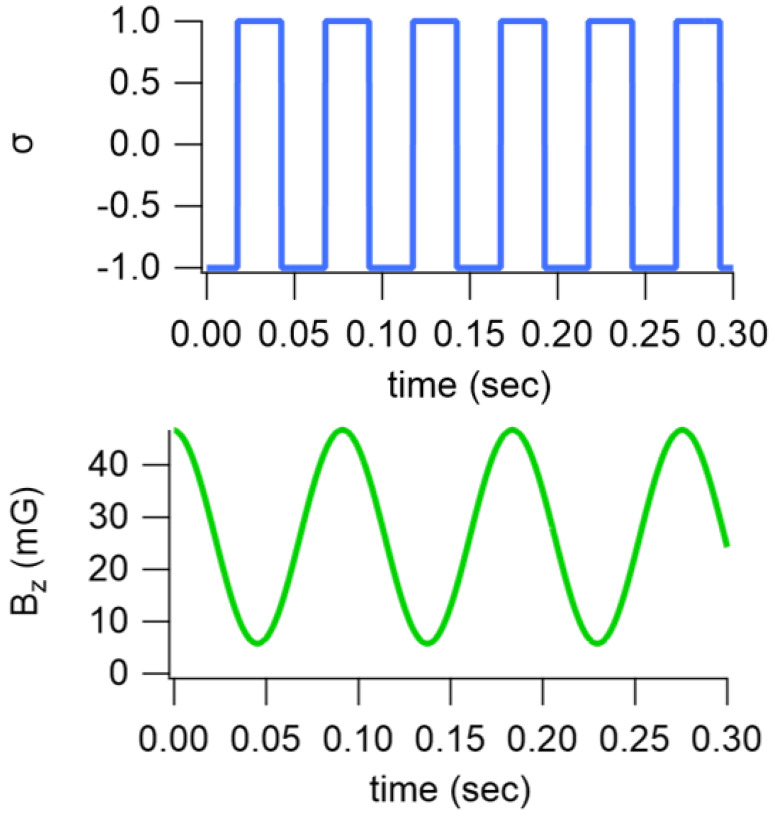
Applied modulation waveforms. (**Top**): the pump light polarization is modulated as a square wave; (**bottom**): the bias field is modulated sinusoidally with a DC offset.

**Figure 4 sensors-23-04649-f004:**
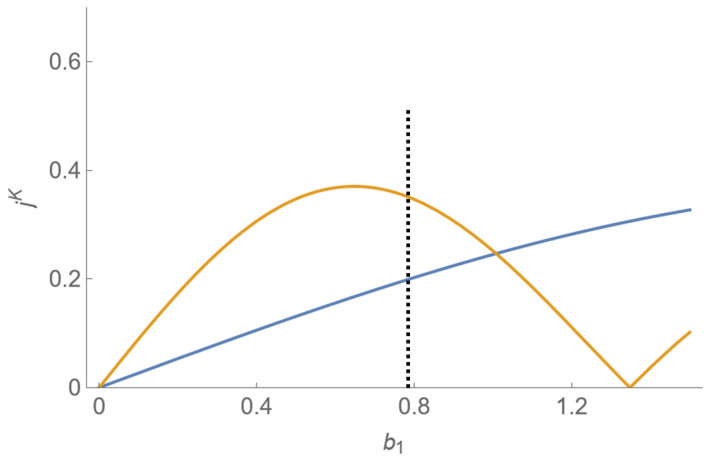
Amplitude coefficient (*j*) vs. PDM modulation index ($b_{1}$) for ${}^{129}$Xe (orange) and ${}^{131}$Xe (blue), given a drive scheme $\omega_{d}^{a}=\omega_{PM}+\omega_{PDM}$, $\omega_{d}^{b}=\omega_{PM}-\omega_{PDM}$. The vertical dashed line shows the choice of $b_{1}$ used in this paper.

**Figure 5 sensors-23-04649-f005:**
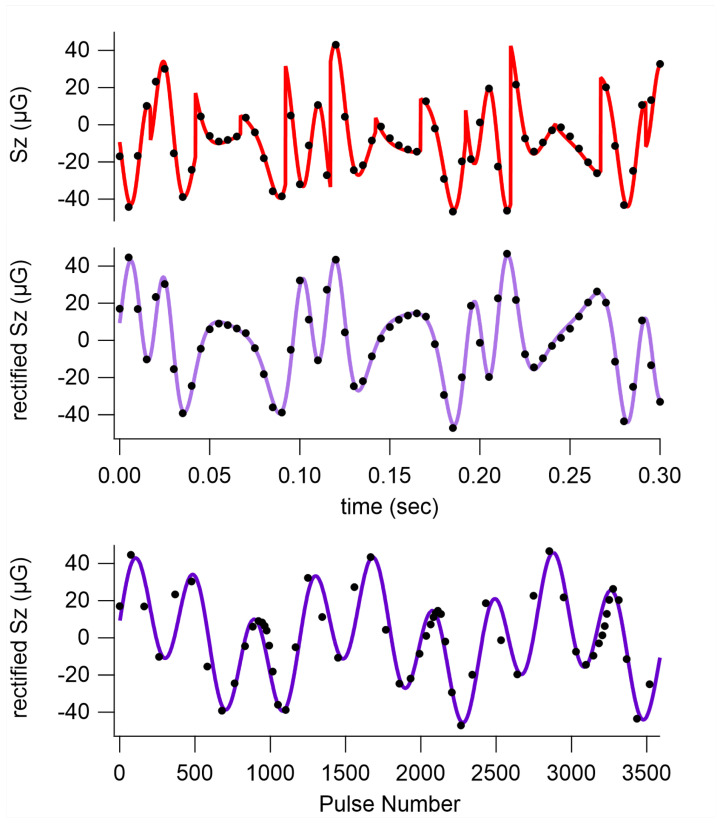
Measured (dots) and theoretical (lines) $S_{z}$ measurements, converted to field units. (**Top**): raw signal vs. time. (**Middle**): rectified signal vs. time; (**bottom**): rectified signal vs. pulse number.

**Figure 6 sensors-23-04649-f006:**
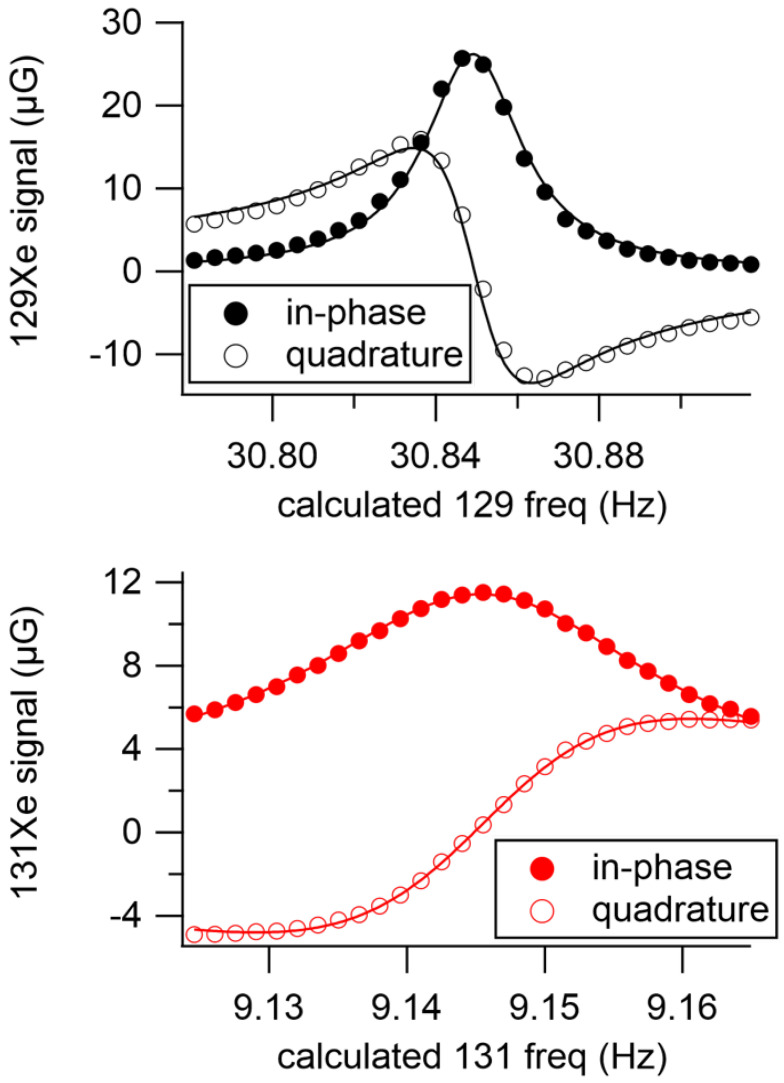
Simultaneous measurements of the Xe linewidths using hybrid drive. (**Top**): ${}^{129}$Xe linewidth = 14 mHz; (**bottom**): ${}^{131}$Xe linewidth = 16 mHz. In-phase and quadrature signals were acquired from the measured magnitude and phase.

**Figure 7 sensors-23-04649-f007:**
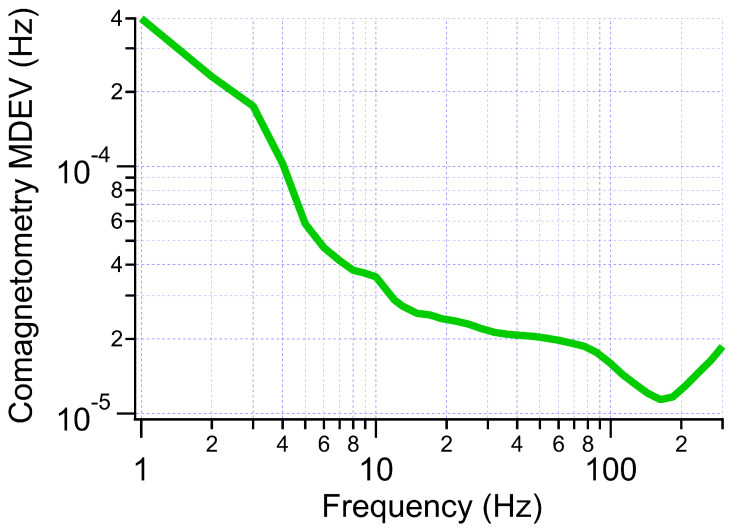
MDEV of comagnetometry calculation using open loop Xe measurements.

**Figure 8 sensors-23-04649-f008:**
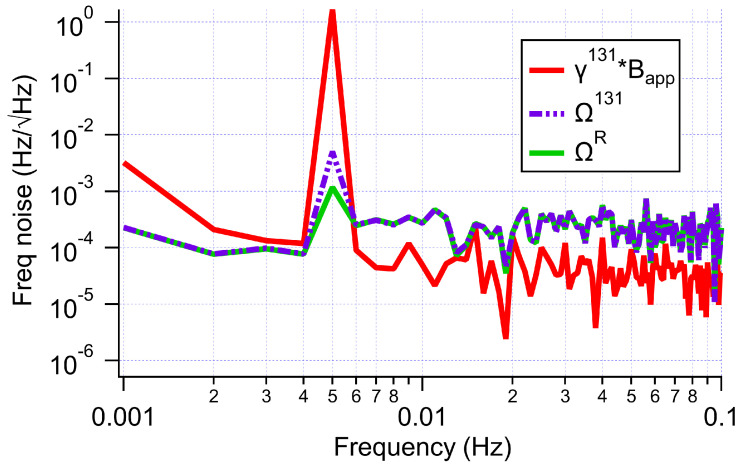
Suppression of the effect of an applied 5 mHz z-field (red) due to field feedback (purple) and comagnetometry (green).

**Figure 9 sensors-23-04649-f009:**
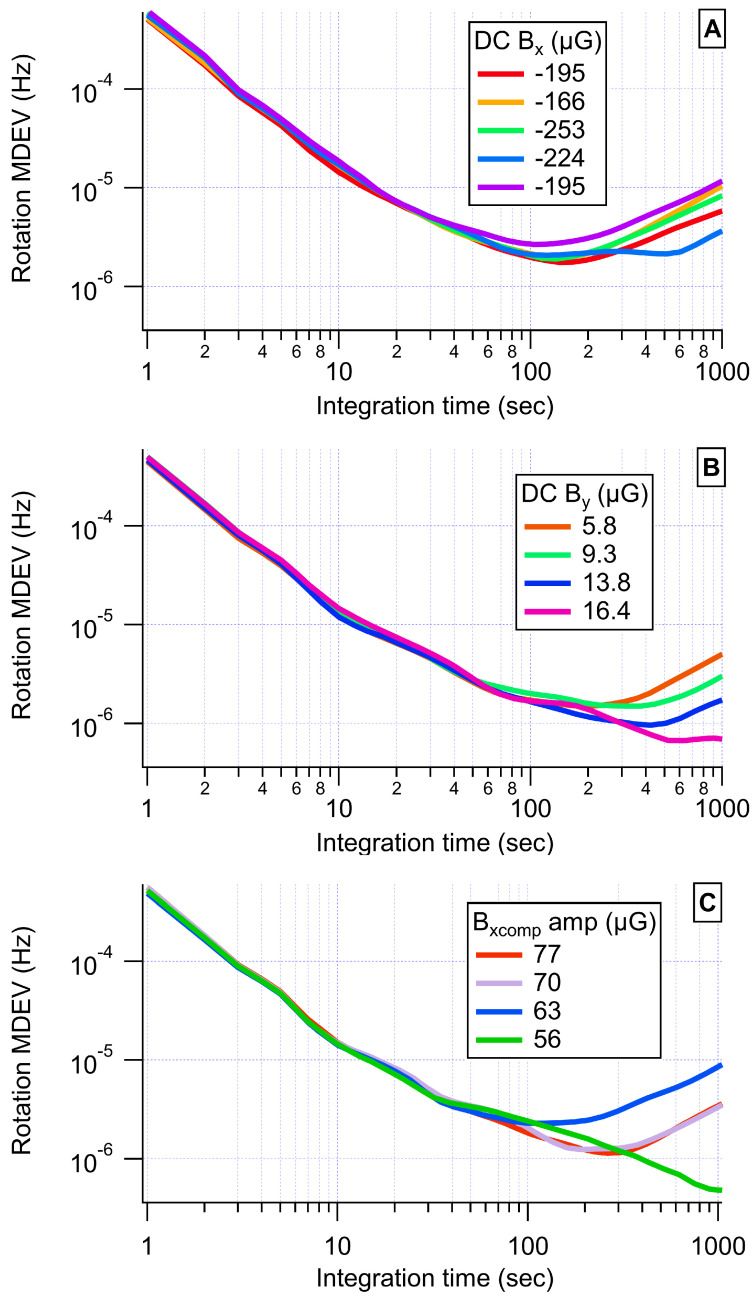
Rotation MDEVs for transverse field scans. We compared MDEVs to find the lowest bias instability for scans of DC $B_{x}$ (**A**), DC $B_{y}$ (**B**), and the AC x-compensation amplitude (**C**). We observed some drift over time, as is evident for DC $B_{x}=-195\;\mu$G.

**Figure 10 sensors-23-04649-f010:**
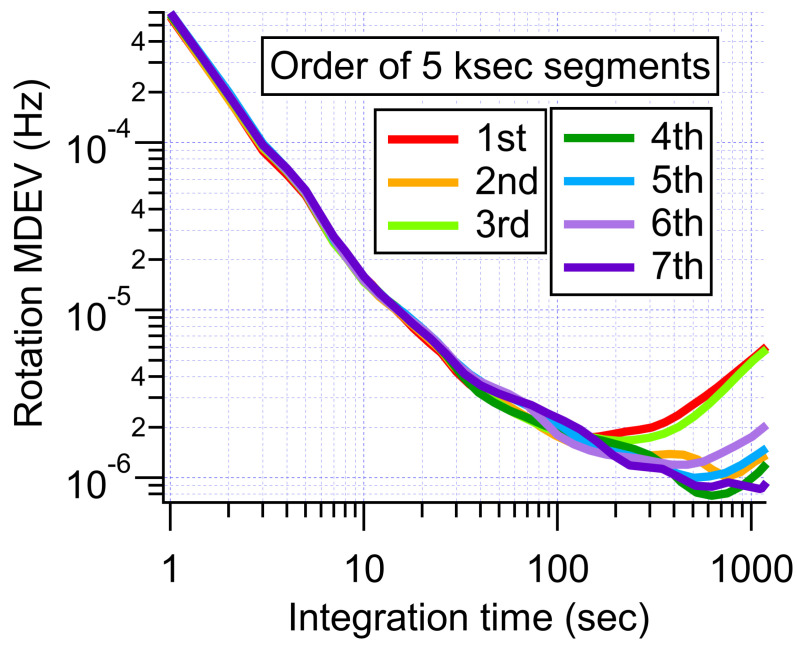
MDEVs of the single 10-hour long rotation measurement cut into 5 ks segments.

**Figure 11 sensors-23-04649-f011:**
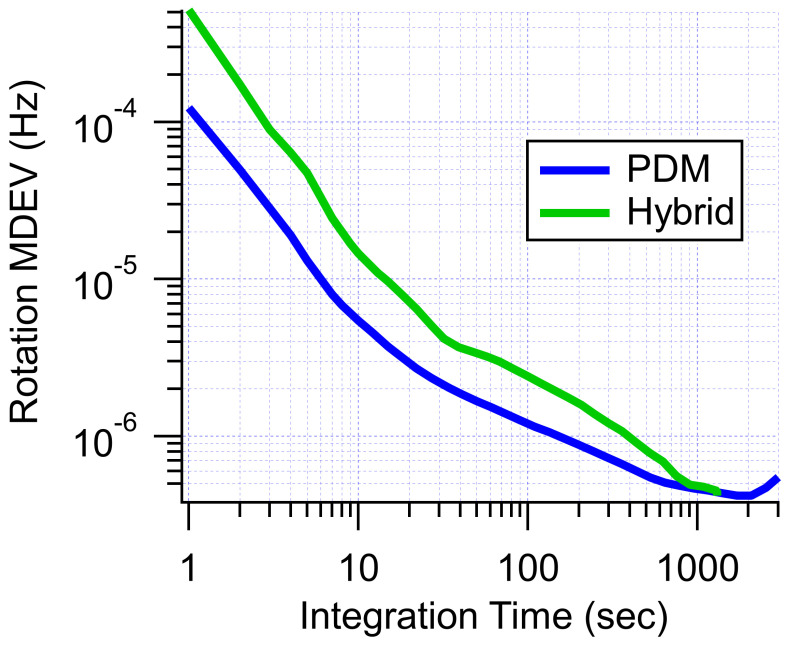
MDEVs for best-to-date stability results for PDM (blue) and hybrid (green) drive.

**Figure 12 sensors-23-04649-f012:**
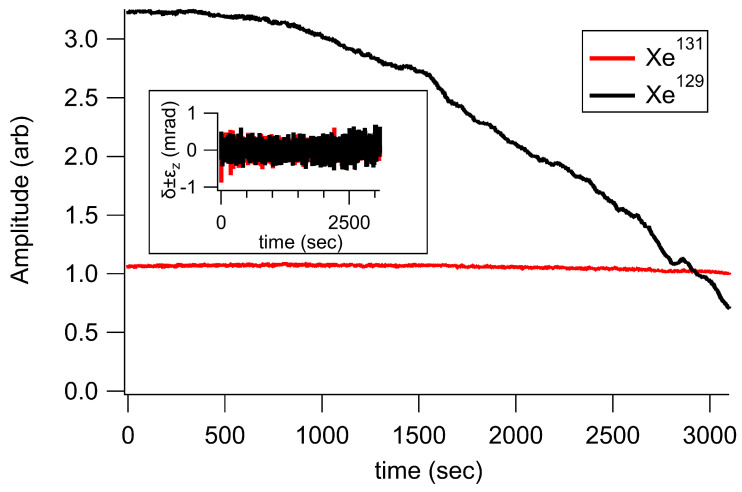
An example of magnitude drift when applying DSDO feedback. The amplitude (arbitrary units) drifted dramatically over the course of several thousand seconds (especially for ${}^{129}$Xe—black), while the measured phase shifts (inset) were held to zero. This implies the feedback is driving Xe further and further off resonance. A low pass filter at 0.1 Hz was applied to the shown phase shifts.

## Data Availability

The data presented in this manuscript are publicly available in a Dryad repository at https://doi.org/10.5061/dryad.ksn02v78p (accessed on 1 March 2023).

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
