# Peer review of "Combined Polarization/Magnetic Modulation of a Transverse NMR Gyroscope"

_sensors, 2023, doi:10.3390/s23104649_

Round 1

Reviewer 2 Report

In the manuscript entitled “Combined Polarization/Magnetic Modulation of a Transverse NMR Gyroscope”, the authors describe the fundamentals of operation for a hybrid-driven NMR gyroscope, how the Xe and Rb polarizations are obtained after the use of the Bloch equations and the comagnetometry using different conditions. Overall, this paper is easy to read and follow. However, although the motivation of such analysis is presented in detail in section 1, I find problems to understand how the proposed methodology could be implemented and if it has been already validated elsewhere. I know that in section 2, a device is discussed, but I do not know if it is a concept or an already running system. In the last case is true, the brand  of principal elements in the system should be added.

I think that this is a good tutorial regarding the application of methods proposed by authors in the processing of data and planning of an experiment, but it lacks of the consideration of operational aspects, for example, sources of uncertainty.

In addition Figure 1 is of low-quality and there is no conclusions section.

I believe this manuscript can be published after these comments are considered.
